# Proof of Concept Testing of Safe Patient Handling Intervention Using Wearable Sensor Technology

**DOI:** 10.3390/s23125769

**Published:** 2023-06-20

**Authors:** Michael Callihan, Brylan Somers, Dhruv Dinesh, Lauren Aldred, Kaitlyn Clamp, Alyssa Treglown, Cole Custred, Kathryn Porteous, Emily Szukala

**Affiliations:** Capstone College of Nursing, The University of Alabama, Tuscaloosa, AL 35401, USA; bssomers@crimson.ua.edu (B.S.); ddinesh@crimson.ua.edu (D.D.); lealdred@crimson.ua.edu (L.A.); kmclamp@crimson.ua.edu (K.C.); aatreglown@crimson.ua.edu (A.T.); ccustred@crimson.ua.edu (C.C.); kaporteous@crimson.ua.edu (K.P.); easzukala@crimson.ua.edu (E.S.)

**Keywords:** healthcare worker, injury prevention, motion capture, intervention

## Abstract

Background: Healthcare workers make up one of the occupations in the United States that experience the most musculoskeletal injuries. These injuries are often related to the movement and repositioning of patients. Despite previous injury prevention attempts, injury rates remain at an unsustainable level. The purpose of this proof-of-concept study is to provide preliminary testing of the impact of a lifting intervention on common biomechanical risk factors for injury during high-risk patient movements.; Methods: A before-and-after (quasi-experimental) design was utilized to compare biomechanical risk factors before and after a lifting intervention. Kinematic data were collected using the Xsens motion capture system, while muscle activations were collected with the Delsys Trigno EMG system. Results: Improvements were noted in the lever arm distance, trunk velocity, and muscle activations during the movements following the intervention; Conclusions: The contextual lifting intervention shows a positive impact on the biomechanical risk factors for musculoskeletal injury among healthcare workers without increasing the biomechanical risk. A larger, prospective study is needed to determine the intervention’s ability to reduce injuries among healthcare workers.

## 1. Introduction

Healthcare workers are consistently among the occupations that experience the highest number of musculoskeletal injuries in the United States, often disproportionate to those seen in general industry [1,2]. These numbers remain high despite the large amount of work done to implement mechanical lifting devices [3,4,5,6], zero lift policies [7,8], and educational offerings on safe patient handling [9,10]. Further research is needed to address musculoskeletal injuries among nurses [3,11]. This research must be targeted at improving the movements and positioning of the nurse [12] while providing methods to limit the amount of weight the healthcare worker must manually move.

Nurses are highly susceptible to back injuries related to the micro-trauma of repeated lifting and moving patients, spending a lot of time walking, bending, standing, and functioning in awkward positions [13]. The education offered in nursing schools requires improvement to address outdated information and must incorporate current best practices to include the proper utilization of mechanical lifts while adopting appropriate body positioning to limit the risk of injury [14]. Often, teaching methods are vague in nature or have poor form demonstrated during the techniques [15,16]. The notion of lifting with your legs instead of your back is often taught but negated by the positioning of the bed at hip height for the lifter. Direct biomechanical evaluation of common nursing movements and potential teaching interventions will allow for improvements in the current teaching methods.

Overexertion has been identified as a leading cause of back injuries among healthcare workers [17], which led to the development of mechanical lifting devices to decrease the load the healthcare worker is required to manually lift. While the use of mechanical lifting devices is ideal for reducing this load, their use in an emergency setting is not practical for patient care. The patient must be rolled on their side and the lifting device placed requiring the healthcare worker to manually log roll the patient. This movement is a strain on the nurse physically and is not even possible on the emergency medical services (EMS) stretcher the patient arrives on due to the narrow footprint of the elevated stretcher. This lack of ability to utilize the mechanical lifting device requires the healthcare workers to manually transfer the patient to the other bed. The log roll and the horizontal transfer of a patient have both previously been identified as high-risk movements for the healthcare worker [18,19,20,21] yet they are necessary when providing emergency medical care.

Muscle fatigue often leads to changes in the movement patterns of a worker [22]. Changes to the angles of the lower back [23,24,25], the distance between the load and the spine (lever arm distance) [25], peak velocity of the trunk during a movement [26], and valgus positioning of the knee [27,28,29] have been identified as biomechanical factors that may increase the risk of injury, especially when used in repetitive motions. Traditional weightlifting training techniques include controlling these body positions while directing muscle activation to protect the spine through core stabilization muscles [30,31].

Aside from the health of the healthcare worker performing the lifting and moving tasks, every interaction must consider the safety of the patient. Medical errors are estimated to be a leading contributor to death in the United States [32]. The wellness of the healthcare worker is critical to limiting the likelihood of committing a medical error [33]. Overexertion and fatigue, which are both effected by poor body mechanics, have been identified as contributing factors to medical errors in healthcare [34]. The development of a safer lifting intervention will help limit musculoskeletal injuries among practicing healthcare workers, and indirectly improve patient outcomes related to medical error commission.

### Intervention Development

The log roll and horizontal transfer of a patient have been identified as high-risk movements [18,19] that, despite the risk, are essential in nursing. The performance of these movements while in an awkward posture account for as much as 80% of the stress experienced on the lower back [12,35]. Unfortunately, the repeated nature of these movements often leads to the development of muscle memory for the movement, which must be changed in order to improve nurse outcomes [36]. The intervention is an expansion of previous work that demonstrated a reduction in the lever arm distance during the lifting and moving of a simulated patient [37] to examine trunk velocity, knee angles, and muscle activation during the movements.

The intervention employed a simple approach to examine some common variables in the lifting and moving processes [38]. Core stabilization is a foundational component of any strength training program [31,39,40]. The core stabilizing sequence utilized in the intervention is based on previous work related to weightlifting programming [30]. By having the sheet extended further from the patient, with knees bent, feet shoulder width apart, and an underhand grip, the worker can achieve natural positioning throughout the movement. The purpose of this proof-of-concept study was to test a lifting intervention for improvements on trunk velocity, trunk angle, valgus knee positioning, and muscle activation of the core during the log roll and horizontal transfer of a patient.

## 2. Materials and Methods

### 2.1. Study Design

This study was approved by the University of Alabama IRB (protocol #21-03-4434) prior to beginning collections. The study was designed as a before-and-after (quasi-experimental) proof of concept to determine if an improvement was noted in the angles of the lower back, velocity of the trunk, activation of core stabilizing muscles, and angles of the knee based on coaching for improved lifting techniques. A before and after model was utilized with participants using the teaching provided in their Fundamentals of Nursing course textbook (Table 1) [15,16], receiving the intervention training developed by the research team (Table 2), and then performing the lifting tasks again. The movements tested were the log roll of the patient and the horizontal transfer, both of which have been identified as high-risk patient movements [18,19]. Due to the high-risk nature of these movements, a proof-of-concept method was utilized to determine if a change in biomechanical risk factors could be noted based on the lifting intervention. This approach allowed for the safety of all participants while testing a new intervention strategy that expanded on previous work by the primary investigator [37].

### 2.2. Setting

Participants were recruited from the college of nursing within the PI’s home institution and asked to come to the clinical practice laboratory at the college. In the private laboratory setting, participants completed each movement for three repetitions under each condition: textbook training and intervention training. During the movement testing, two emergency department stretchers were placed side by side, with a member of the research team weighing approximately 75 kg serving as the patient.

### 2.3. Participants

Participants included healthy nursing students from the institution that had finished the second semester of the nursing program. Inclusion criteria were ages 18–30 and having completed the initial lifting and movement training in the nursing program of study. The exclusion criteria was a history of musculoskeletal injury. Given the risk of injury noted in patient movement and handling, a small, healthy sample was utilized for the proof of concept to determine if the intervention produced a change in biomechanical factors.

### 2.4. Intervention

The lifting movements were coached by a member of the research team, with the textbook method (Table 1) being coached first and the intervention method (Table 2) being coached immediately following the initial completion of the movements. Members of the research team were trained and approved for the coaching of the lifting techniques by the primary investigator, who is certified in weightlifting and preventative lifting techniques, and then supervised during the delivery of the intervention by the PI to ensure accuracy in the training. Coaching of the movements took place at the bedside, with hands on demonstrations by the researcher followed by demonstrations by the participant. Once the participant was comfortable with the movements as coached, data were collected during the movement. To ensure consistency in the teaching methods, a printed form containing Table 1 and Table 2 was used by the researcher giving the training. The coaching of each lifting method, including hands-on practice, took approximately 10 min for each participant (8.5–11 min).

### 2.5. Variables of Interest

Peak and average angles of the lower back, lever arm distance, and velocity of the lumbar spine, were collected and processed. Following the collection of the movements, the data were processed in the Xsens motion capture software, with the joint angles calculated based on the orientation of the distal body segment to the proximal body segment. The data were further processed through Visual 3D software version 2021x 64 (C Motion, Inc., Germantown, MD, USA), with raw data utilized to calculate peak and average data. The lever arm distance was calculated by creating a virtual marker at the level of the lumbosacral joint that tracked with the right hand.

Peak muscle activation data were collected through EMG within the Xsens motion capture software and extracted as Root Mean Square (RMS) values of the amplitude. Muscle activations of interest included the erector spinae, rectus abdominus, external oblique, trapezius, rear deltoid, gluteus maximus, rectus femoris, and biceps femoris.

All data were analyzed in spreadsheet format. Given the small sample size, statistical analysis was not possible; however, descriptive statistics were obtained through Excel spreadsheet analysis and the effect size calculated from the intervention to determine needed sample sizes for future studies using SPSS version 29 (IBM, Armonk, NY, USA).

#### 2.5.1. Fatigue Angle of Lumbar Spine and Trunk Velocity

Healthcare workers frequently are in awkward positions while performing lifting and moving tasks [12,21]. Previous work has identified that the tissue of the lumbar spine fails more rapidly when in a flexed position of 22.5 and 45 degrees [23,24]. The peak flexion of the lumbar spine was measured as the relationship between the pelvis and the trunk for this study to determine if the fatigue angles were exceeded, and if exceeded, raw data will be examined to determine the time spent during the movement in this position. The velocity of the trunk has been shown to have a link to lower back pain [41] and was measured during this study.

#### 2.5.2. Lever Arm Distance

Repetitive motion injuries are common in general industry and must be explored in the healthcare setting as well. The Lifting Fatigue Failure Tool (LiFFT) [25] was developed to relate the weight of the load being moved, the maximal horizontal distance from the weight to the spine, and the number of repetitions of the movement throughout the workday. For this study, the lever arm distance was calculated by creating a virtual marker at the level of the lumbosacral joint that tracked with the right hand throughout the movement, similar to previous work by the primary investigator [37].

#### 2.5.3. Muscle Activation

Activation of the core muscles during lifting has been shown to be an important factor in providing an increased level of protection to the lumbar spine during lifting and moving tasks [39,42,43]. While most lifting strategies state to lift with the legs and not the back [44], teaching strategies to do so are limited. Muscle activations were measured for this study using the Deslsys Trigno EMG system.

### 2.6. Data Sources/Measurement

#### 2.6.1. Xsens

Motion capture technology has been utilized as the gold standard for the collection of kinematic data [20] and was used for this study to correlate and validate the angles of the lower back as well as to calculate the angle velocities of the trunk. A commercially available motion capture system, Xsens, utilized seventeen inertial measurement units (IMUs) placed on the body to identify the body segment orientation and the angle of the joints. Following placement of the 17 IMU devices, calibration was completed per manufacturer’s recommendation with the participant standing in the N-pose (upright position, head forward, arms resting at sides), walking forward, and then returning to the initial starting point.

It is known that IMU based motion data is less reliable during the abduction/adduction (*y*-axis) movements of the knees; however, the flexion and extension (*x*-axis) motions have been found to be relatively accurate [45,46]. For this study, the Xsens system was deemed adequate in tracking the motions for each participant in the flexion and extension movements but was not used in tracking the abduction/adduction motions [47,48,49] Further determination of improvements in these motions will require validation using a camera-based system. The IMU based sensors allowed for the collection of motion data in the presence of the hospital beds, which would impede the detection of motion with a camera-based motion capture system. The portability and the ability to track the movements accurately during the movements led the research team to deem this technology appropriate for the current study. Future studies will need to be completed using camera-based technology to provide more accurate results.

#### 2.6.2. Delsys Trigno System

The participant’s skin was prepared for electrode placement by shaving any hair, lightly abrading the surface with sandpaper, then wiping away excess skin oil using an alcohol swab and allowing it to air dry. Electrodes were then applied following the guidelines outlined in the Surface Electrodes for the Non-Invasive Assessment of Muscles (SENIAM) protocol [50,51]. Electromyography (EMG) data was measured on the right side of each participant for the gluteus maximus (GMAX), rectus femoris (RF), biceps femoris (BF), external oblique (EO), rectus abdominus (RA) and erector spinaes (ES), rear deltoid (RD), and medial trapezius (TZ) using surface electrodes. The EMG activity was captured using eight Delsys Trigno sensors (Delsys, Inc., Natick, MA, USA) and synchronized through the Xsens system for synchronization. Bipolar AG surface electrodes (Delsys Inc., Boston, MA, USA) with a fixed inter-electrode distance of 10 mm and a dimension of 10 × 1 mm was used.

### 2.7. Bias

To control for bias in the data collection, participants were coached to perform the patient movements directly from the textbook for the pre-test collection, then were coached from the intervention for the post-test collection. A printed copy of the instructions was utilized by the research team to ensure consistency in the training.

## 3. Results

### 3.1. Participants

Participants were recruited from the College of Nursing at the university. Four participants completed the study, the majority being female (75%) with an average age of 21.7 years (21–22) and a height of 172 cm (166–185 cm). All participants had received the initial safe patient moving and handling training provided at the college and were coached to perform the movements per textbook recommendations.

### 3.2. Log Roll

Composite data for the log roll movement is presented in Table 3. The data indicates a reduction in the maximum lever arm distance for three of the four participants and a reduction in the average lever arm distance for all participants. The peak velocity of the spine increased in three of four participants. On average, an increase in muscle activity was noted in all muscle groups except the external oblique, which demonstrated a decrease of 1.31 mV.

### 3.3. Push Movement

Composite data for the push portion of the horizontal transfer is presented in Table 4. The data indicates a reduction in the peak lever arm distance among three of the four participants and a reduction in the average lever arm distance in two of the four participants. Peak velocity of the spine was noted among two of the four participants. The averaged muscle activity demonstrated an increase in activation of the posterior deltoid, gluteus maximus, and biceps femoris. A reduction in muscle activation was noted in the latissimus dorsi, erector spinae, rectus femoris, rectus abdominus, and external oblique.

### 3.4. Pull Movement

Composite data for the pull portion of the horizontal transfer is presented in Table 5. The data indicates an increase in lever arm maximum and average for three of the four participants. Peak velocity of the spine was also noted in three of the four participants. The average muscle activity increased for the latissimus dorsi, erector spinae, gluteus maximus, biceps femoris, and rectus femoris, with a reduction noted in the posterior deltoid, rectus abdominus, and external oblique.

## 4. Discussion

The log roll and horizontal transfer of a patient are high-risk movements [18,19] that often are necessary for emergency healthcare workers to perform when moving a patient. The safest way to perform this movement is to use a mechanical lifting device to perform the transfer; however, placing the lifting harness under the patient while they are on the EMS stretcher is not safe. The EMS stretcher has a narrow base and does not allow room to safely log roll the patient to place the harness without causing the stretcher to turn over. This makes current safe-lifting equipment unsafe to use for this initial emergency transfer, so the development of safer lifting strategies utilizing proper body mechanics is necessary to help reduce the occurrence of musculoskeletal injuries within this workforce.

Lever arm distance was improved in the log roll and push portions of the horizontal transfer, while a slight (2 cm) increase was noted in the pull portion. A combination of lever arm distance, weight being moved, and number of repetitions performed has been shown to be predictive of lower back injuries [25]. While the weight of the patient and the number of times the healthcare worker will have to perform the movement are unknown, the limiting of the lever arm distance is within the control of the worker. The lengthening of the draw sheet allows the worker to activate their posterior chain (glute, BF, and ES) in the squat position while reducing or maintaining the lever arm distance.

The lowering of the bed to the mid-thigh position of the shortest member of the team allowed for an increase in posterior chain activation for each movement. This change in position accounted for an increase in the ability of the participants to use their legs to perform the movement rather than relying on their backs and arms to produce the force needed to perform the movement. The positioning of the hands in an underhand grip rather than the typical overhand grip allowed the participants to keep their hands below the level of their shoulders, reducing the force experienced [52,53].

The log roll movement showed an increase in trunk velocity of 8 cm per second on average, while the push and pull movements of the horizontal transfer demonstrated a reduction of 10 and 25 cm/s respectively. This increased trunk velocity is not of concern for the safety of the movement when viewed together with the increased activation of the posterior chain (BF, glute, and ES) indicating that the participant was successfully able to lift with their legs rather than their back for each movement. The increased activation of the rectus abdominus would indicate that the participants were able to improve their core stabilization during the movement, potentially improving their intraabdominal pressure, providing better stabilization of the lumbar spine [42,54,55].

The safety of the patient throughout the movement is of utmost importance with any patient transfer. Patients are at risk for skin tears from rough handling [56] which must be accounted for during patient movements. The use of a mechanical lifting device [14,57] and friction reducing slide boards [12], are not always available or practical for use in an emergency setting, making manual lifting a necessity for patient care. While performing the movements before and after the intervention, the patient was lifted more during the intervention, resulting in an expected decrease in friction between the bed and the patient. This decreased friction is hypothesized to decrease the risk of a skin tear secondary to the movement.

### Limitations

Limitations for this proof-of-concept study include the potential bias of the participants, as they potentially exaggerated their movements for the study. This was addressed by the research team, which guided the movements based on the textbook used in their Fundamentals of Nursing course. A second limitation is the small sample size of the collections. This was designed into the study to control for injury risks. Patient movement has been found to be a high-risk activity among healthcare workers. The controlled lab-based testing helped limit the risk for the participants while developing the lifting intervention. The proof-of-concept design allowed for testing of the intervention in a small, controlled group prior to subjecting a larger sample size to the intervention. Future studies will prospectively look at nurses and their reported incidence of musculoskeletal pain, injury, or discomfort.

## 5. Conclusions

Based on the findings of this proof-of-concept study, the lifting intervention demonstrated improvements in the lever arm distance, trunk velocity, and activation of the core stabilization muscles while not increasing the risk of injury related to poor body mechanics. This gives promise to the safety of testing this intervention towards injury prevention among healthcare workers. Work needs to continue to strengthen the rear deltoids and latissimus dorsi of the workers to further reduce the strain of the movements on them. Future implementation must have a greater focus on improving this positioning while performing the movements. 

Future directions for this project include more in-depth testing of the efficacy of the intervention as well as prospective studies examining the impact of the intervention on injury rates among healthcare workers. While biomechanical risk factors have been targeted with this intervention, the clinical results of injury reduction have not been tested and require further work. 

## Figures and Tables

**Table 1 sensors-23-05769-t001:** Textbook directions for log roll and horizontal transfer of a patient.

Movement	Standard Instructions-Patient Will Begin with 2 Draw Sheets in Place
Log Roll	Step 1: Maintain C-spine stabilization if indicated (will not be part of this study) Step 2: The patient is positioned with their legs extended in the usual manner and arms extended by their sides, with palms facing inward. The patient will be gently rolled up onto one arm, to provide both splinting for the body and proper spacing for the movement. Step 3: The caregiver reaches across the patient and securely holds the shoulder and the hip, ensuring the far arm remains in place. With the sheet rolled tight to the patient’s body, grasp the sheet at the level of the shoulder and the hip. Step 4: When ready, roll the patient to their side.
Horizontal Transfer of Patient (Push and Pull)	Step 1: Position the participant to the side of the bed that the patient will be moving to and another nurse on the opposite side. Step 2: fan fold the sheet on both sides against patient Step 3: Stand with feet spread widely with one foot slightly in front of the other and grasp the draw sheet. Step 4: on count of three, pull the patient to the new desired position. (slide board is recommended)

**Table 2 sensors-23-05769-t002:** Intervention instructions for safe moving.

Movement	Intervention Instructions—Start with 2 Draw Sheets under the Patient
Core Activation Exercise [29]	Core activation must be utilized throughout each of the movement patterns. To practice this skill, participants are instructed to stand up straight with feet shoulder width apart, shoulders back, and chest pushed forward. Then, they contract gluteal muscles fully and take a deep breath. On the exhale of the breath, they contract the abdominal muscles, and while maintaining contraction of the abdominal muscles, relax the gluteal muscles.
Log Roll of Patient	With the nurse positioned on one side of the patient, position the bed at a mid-thigh position for the nurse. The second nurse will be positioned on the other side of the patient. The nurse will reach across the patient and pull the draw sheet across the patient. With an underhand grip, the nurse will assume the squat position with the arms extended holding the sheet. On a count of three, the nurse will pull the sheet toward their chest as they stand up. The nurse on the other side of the patient will assume the squat position with hands placed in an underhand grip at the hip and shoulder of the patient. At the count of three, the nurse will drive through their heels and lift the patient to their side.
Horizontal Transfer of Patient (Push and Pull)	Team members (2) will be positioned on either side of the patient, who is placed on a slide board. The bed will be adjusted to allow each team member to grip the draw sheet beneath the patient, maintaining slightly bent knees and a neutral spinal position. In a synchronized manner, the team member on the pushing side will push through their heel, extending their arms while guiding the patient across the slide board. The receiving team member will push through their heels, stand upright, retract their elbows, and squeeze their shoulder blades together to assist in smoothly receiving the patient A third nurse will be positioned to support the patient in the side lying position

**Table 3 sensors-23-05769-t003:** Log roll data for Lever Arm (LA) maximum and average, trunk velocity, trunk flexion, Posterior Deltoid (PostDelt), Latissimus Dorsi (LatDorsi), Erector Spinae (ES), Gluteus Maximus (Glute), Biceps Femoris (BF), Rectus Femoris (RF), Rectus Abdominis (RA), and External Oblique (EO). Condition 1 is pre-intervention; Condition 2 is post-intervention; and diff 1–2 is the change between conditions.

Part	Con	LA Max	LA Avg	Velocity	Flexion	PostDelt	Lat Dorsi	ES	Glute	BF	RF	RA	EO
1	1	0.66 (0.13)	0.44 (0.12)	−0.12 (0.03)	−1.26 (1.74)	−0.10 (0.01)	−0.33 (0.03)	−0.07 (0.01)	0.76 (0.13)	0.20 (0.04)	−0.11 (0.01)	−0.36 (0.07)	−0.03 (0.00)
1	2	0.65 (0.18)	0.34 (0.19)	0.10 (0.04)	0.00 (2.04)	0.33 (0.04)	−0.36 (0.05)	−0.12 (0.02)	0.99 (0.12)	0.30 (0.06)	−0.14 (0.02)	−0.63 (0.08)	0.45 (0.03)
1–2		0.01	0.10	0.01	1.26	−0.23	−0.04	−0.05	−0.23	−0.11	−0.03	−0.28	−0.15
2	1	0.64 (0.12)	0.43 (0.10)	−0.12 (0.05)	−5.83 (3.93)	−0.23 (0.02)	−0.69 (0.17)	−5.49 (1.27)	−0.28 (0.07)	−0.18 (0.03)	0.05 (0.01)	−0.13 (0.02)	−0.03 (0.00)
2	2	0.56 (0.14)	0.25 (0.13)	0.24 (0.11)	0.00 (3.95)	−0.31 (0.03)	1.80 (0.21)	−5.49 (0.88)	0.60 (0.13)	−0.18 (0.02)	0.07 (0.01)	−0.17 (0.04)	−0.03 (0.00)
1–2		0.08	0.18	−0.12	5.83	−0.08	−1.11	0.00	−0.32	0.00	−0.03	−0.04	0.00
3	1	0.65 (0.09)	0.40 (0.10)	−0.12 (0.05)	−9.90 (5.16)	−0.24 (0.02)	−0.27 (0.04)	−0.18 (0.02)	0.63 (0.15)	−0.24 (0.06)	−0.04 (0.01)	−0.20 (0.05)	−5.50 (1.67)
3	2	0.68 (0.21)	0.30 (0.21)	−0.16 (0.08)	−8.69 (5.17)	0.10 (0.02)	−0.63 (0.06)	0.22 (0.05)	0.90 (0.14)	−0.37 (0.06)	−0.05 (0.01)	0.33 (0.07)	−0.07 (0.01)
1–2		−0.03	0.10	−0.05	1.22	0.14	−0.36	−0.04	−0.27	−0.13	−0.01	−0.13	5.43
4	1	0.61 (0.11)	0.39 (0.12)	0.05 (0.03)	0.00 (3.71)	0.67 (0.11)	−0.36 (0.05)	−0.15 (0.02)	−0.03 (0.01)	−0.10 (0.01)	−0.08 (0.01)	−0.07 (0.01)	−0.08 (0.01)
4	2	0.53 (0.13)	0.30 (0.12)	0.26 (0.09)	−5.93 (3.81)	0.95 (0.21)	−0.22 (0.04)	−0.16 (0.03)	−0.05 (0.01)	−0.14 (0.03)	−0.08 (0.02)	0.08 (0.02)	−0.08 (0.02)
1–2		0.08	0.08	−0.21	5.93	−0.10	0.14	−0.01	−0.02	−0.04	−0.01	−0.01	0.00
avg	1	0.64	0.42	−0.11	−4.04	−0.26	−0.41	−1.47	0.42	−0.17	−0.06	−0.19	−1.40
avg	2	0.60	0.30	0.19	−1.07	0.34	−0.68	−1.49	0.63	−0.24	−0.09	−0.30	−0.09
1–2		0.03	0.12	−0.08	2.97	−0.08	−0.26	−0.02	−0.22	−0.08	−0.03	−0.11	1.31

**Table 4 sensors-23-05769-t004:** Push movement data for Lever Arm (LA) maximum and average, trunk velocity, trunk flexion, the Posterior Deltoid (PostDelt), Latissimus Dorsi (LatDorsi), Erector Spinae (ES), Gluteus Maximus (Glute), Biceps Femoris (BF), Rectus Femoris (RF), Rectus Abdominis (RA), and External Oblique (EO). Condition 1 is pre-intervention; Condition 2 is post-intervention; and diff 1–2 is the change between conditions.

Part	Con	LA Max	LA Avg	Velocity	Flexion	PostDelt	Lat Dorsi	ES	Glute	BF	RF	RA	EO
1	1	0.70 (0.19)	0.50 (0.18)	−0.15 (0.05)	−0.99 (2.87)	−0.15 (0.02)	−0.53 (0.06)	−0.14 (0.02)	0.95 (0.13)	0.38 (0.06)	−0.14 (0.03)	−0.36 (0.05)	−0.16 (0.02)
1	2	0.67 (0.13)	0.38 (0.14)	−0.13 (0.05)	−2.28 (2.95)	0.23 (0.03)	−0.32 (0.05)	−0.12 (0.03)	0.80 (0.08)	0.32 (0.07)	−0.14 (0.03)	−0.36 (0.04)	−0.04 (0.01)
1–2		0.03	0.12	0.02	−1.28	−0.08	0.21	0.02	0.15	0.05	0.00	0.00	0.18
2	1	0.70 (0.14)	0.53 (0.13)	0.36 (0.12)	−3.46 (1.31)	−0.10 (0.02)	−0.75 (0.15)	5.49 (1.15)	0.48 (0.07)	−0.20 (0.04)	−0.05 (0.01)	−0.15 (0.02)	−0.03 (0.00)
2	2	0.61 (0.12)	0.45 (0.11)	0.34 (0.06)	−2.11 (3.56)	−0.30 (0.05)	1.54 (0.14)	−2.12 (0.19)	0.21 (0.04)	−0.26 (0.06)	−0.06 (0.01)	−0.12 (0.03)	−0.03 (0.01)
1–2		0.10	0.07	0.02	1.35	−0.21	−0.80	3.38	0.27	−0.07	−0.01	0.03	0.00
3	1	0.78 (0.22)	0.49 (0.24)	0.19 (0.09)	−9.74 (3.50)	−0.10 (0.01)	−0.51 (0.04)	−0.14 (0.02)	−0.44 (0.07)	−0.50 (0.10)	−0.05 (0.01)	−5.50 (0.58)	0.04 (0.01)
3	2	0.71 (0.15)	0.51 (0.15)	−0.50 (0.15)	−12.7 (2.52)	0.12 (0.01)	−0.40 (0.05)	0.15 (0.03)	0.49 (0.09)	−0.52 (0.12)	−0.06 (0.01)	0.78 (0.12)	−0.06 (0.01)
1–2		0.07	−0.02	−0.31	−3.03	−0.01	0.11	0.00	−0.05	−0.02	−0.01	4.72	−0.02
4	1	0.65 (0.19)	0.41 (0.19)	0.20 (0,05)	0.00 (2.66)	0.65 (0.10)	−0.19 (0.04)	−0.35 (0.05)	−0.06 (0.01)	−0.18 (0.02)	−0.33 (0.02)	−0.58 (0.06)	0.13 (0.02)
4	2	0.66 (0.17)	0.46 (0.14)	0.50 (0.16)	0.00 (3.71)	0.56 (0.08)	0.69 (0.07)	0.51 (0.08)	−0.06 (0.01)	−0.29 (0.05)	0.14 (0.03)	−0.08 (0.01)	−0.12 (0.02)
1–2		−0.01	−0.05	−0.30	0.00	0.10	−0.51	−0.17	−0.01	−0.11	0.19	0.50	0.00
avg	1	0.71	0.48	0.15	−3.55	0.07	−0.49	1.22	0.23	−0.12	−0.14	−1.65	0.09
avg	2	0.66	0.45	0.06	−4.29	0.15	0.38	−0.39	0.36	−0.19	−0.03	0.05	−0.06
1–2		0.05	0.03	0.10	−0.74	−0.08	0.11	0.82	−0.13	−0.06	0.11	1.59	0.03

**Table 5 sensors-23-05769-t005:** Pull movement data for Lever Arm (LA) maximum and average, trunk velocity, trunk flexion, the Posterior Deltoid (PostDelt), Latissimus Dorsi (LatDorsi), Erector Spinae (ES), Gluteus Maximus (Glute), Biceps Femoris (BF), Rectus Femoris (RF), Rectus Abdominis (RA), and External Oblique (EO). Condition 1 is pre-intervention; Condition 2 is post-intervention; and diff 1–2 is the change between conditions.

Part	Con	LA Max	LA Avg	Velocity	Flexion	PostDelt	Lat Dorsi	ES	Glute	BF	RF	RA	EO
1	1	0.68 (0.18)	0.37 (0.17)	−0.16 (0.06)	−3.03 (3.27)	0.24 (0.04)	−1.20 (0.15)	−0.29 (0.08)	1.23 (0.25)	−0.48 (0.08)	0.51 (0.07)	−0.84 (0.15)	−0.12 (0.01)
1	2	0.70 (0.23)	0.41 (0.17)	0.26 (0.06)	−1.40 (1.81)	0.24 (0.03)	−0.80 (0.12)	−0.29 (0.06)	1.23 (0.17)	0.57 (0.08)	0.51 (0.05)	−0.71 (0.16)	0.15 (0.02)
1–2		−0.03	−0.05	−0.10	1.63	0.00	0.40	0.00	−0.20	−0.08	0.00	0.13	−0.04
2	1	0.69 (0.15)	0.44 (0.13)	−0.21 (0.07)	−4.59 (1.51)	0.09 (0.01)	1.00 (0.13)	−1.21 (0.15)	0.58 (0.12)	−0.34 (0.06)	−0.08 (0.01)	−0.11 (0.03)	−0.04 (0.01)
2	2	0.71 (0.21)	0.37 (0.20)	−0.22 (0.09)	0.00 (4.36)	−0.23 (0.02)	−5.49 (3.09)	−5.49 (1.06)	0.66 (0.17)	−0.41 (0.06)	−0.06 (0.01)	−0.14 (0.03)	−0.04 (0.01)
1–2		−0.02	0.07	−0.01	4.59	−0.14	−4.49	−4.28	−0.09	−0.07	0.02	−0.03	0.00
3	1	0.77 (0.23)	0.37 (0.21)	−0.29 (0.10)	−11.9 (5.36)	0.13 (0.02)	−0.68 (0.11)	0.25 (0.05)	1.10 (0.19)	0.58 (0.11)	−0.45 (0.07)	−0.64 (0.11)	−0.42 (0.06)
3	2	0.77 (0.24)	0.50 (0.24)	−0.32 (0.15)	−18.5 (4.13)	0.23 (0.03)	−0.41 (0.06)	−2.51 (0.16)	1.09 (0.18)	−0.50 (0.11)	0.09 (0.02)	0.58 (0.10)	−0.10 (0.01)
1–2		0.00	−0.14	−0.04	−6.51	−0.10	0.28	−0.55	0.00	0.08	0.36	0.06	0.32
4	1	0.62 (0.19)	0.34 (0.19)	−0.37 (0.11)	−1.46 (3.50)	1.44 (0.14)	−0.45 (0.06)	−0.27 (0.05)	−0.07 (0.02)	−0.22 (0.03)	0.37 (0.04)	−0.22 (0.02)	−0.10 (0.02)
4	2	0.65 (0.21)	0.34 (0.19)	0.26 (0.14)	0.00 (1.29)	1.40 (0.25)	−0.45 (0.07)	−0.37 (0.06)	−0.08 (0.02)	−0.34 (0.08)	−0.08 (0.02)	−0.11 (0.01)	−0.09 (0.01)
1–2		−0.03	0.00	0.11	1.46	0.04	0.00	−0.10	−0.01	−0.13	0.29	0.11	0.01
avg		0.69	0.38	−0.26	−5.27	0.48	−0.33	−0.38	0.14	−0.11	0.09	−0.45	−0.17
avg		0.71	0.41	−0.01	−4.97	0.41	−1.79	−1.34	0.73	−0.17	0.12	−0.10	−0.02
1–2		−0.02	−0.03	0.25	0.29	0.07	−1.46	−0.96	−0.59	−0.06	−0.03	0.36	0.15

## Data Availability

Data is available by contacting the corresponding author.

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
