# Peer review of "Proof of Concept Testing of Safe Patient Handling Intervention Using Wearable Sensor Technology"

_sensors, 2023, doi:10.3390/s23125769_

Round 1
Reviewer 1 Report
A novel lifting intervention instruction was proposed aiming to reduce the injury risk of the healthcare worker from the aspects of regulating the patient repositioning movements.. Several biomechanical parameters related to the risk of injuries for workers calculated and compared under textbook directions and the proposed instructions. The author's work is meaningful, but the analysis method is not clear, and the conclusions obtained need to be more cautious and verified with more rigorous and organized method.
Major comments:
1. The proposal of a new medical related guidance should be more rigorous. The author only verified the biomechanical risk factors for injury proposed in previous literatures. However, it is important to note that this movement is associated with patients, so the risk of injury to the patients should also be considered.
2. In this study, method is chaotic, and the description of the method is not clear. More rigorous analysis schemes need to be reconsidered.
For example, (1) section 2.1 and section 2.2 are repetitive in the experimental process. And section 2.4 and section 2.6 are repetitive in the selection and calculation of biomechanical parameters. (2) In section 2.4.1, the authors indicated that the lumbar spine tissues were fails more rapidly at the 22.5 and 45 degrees flexion positions, but the peak lumbar flexion was measured in this study. (3) In the second paragraph of section 2.5.1, the author stated that the motion data based on IMU are unreliable, but author still use IMU to obtain valgus position of the knee. The same paragraph also stated that the research team deemed the Xsens is accurate enough. This viewpoint lacks logic and support. (4) In section 2.6, it was stated that statistical analysis could not be conducted due to the small sample size, however section 2.9 was named Statistical methods. (5) The abstract mentioned that the test-retest model is used. but not descripted in the method. (6) What is the purpose of section 2.8 “Study size”? This part just simply redescribed the experimental procedure and the number of subjects.
3. The number of subjects was limited, and the analysis method is relatively simple, it is difficult to achieve the verification purpose hoped by the author.
Minor comments:
4. It is pointed out in the abstract that the paper used the test-retest model to compare the two movement instructions of patient repositioning, but the related results didn’t appear in the paper, please check whether it was missing. In addition, the test-retest method examines the correlation of parameters under two tests. For this paper, it seems to focus more on the biomechanical differences than the similarities between the two conditions. Please consider whether the test-retest method is appropriate for the study aims.
5. The result description of the second sentence in section 3.3 was repeated:’’ The data indicates a reduction in the peak lever arm distance among 3 of 4 participants and a reduction in the peak lever arm distance in 2 of the 4 participants.”
6. The article has many formatting problems, such as inconsistent fonts, missing punctuation, and incorrect formatting of references.
Language and grammar needs to be carefully checked and corrected.
Author Response
Thank you for reviewing our submission. Our team has tried to incorporate all of your suggestions and answer your questions to the best of our ability.

Reviewer 2 Report
Manuscript ID: sensors-2421760-peer-review-v1
Manuscript title: Proof of Concept Testing of Safe Patient Handling Intervention Using Wearable Sensor Technology
Comments
This manuscript reports a proof-of-concept study designed to test a lifting intervention for improvements on the trunk velocity, trunk angle, valgus knee positioning, and muscle activation of the core during the logroll and horizontal transfer of a patient. The Introduction sections briefly poses the study rationale and context. The study aim is clear. However, the reported study design does not accurately describe the authors’ choice.
Major comments
1. Abstract (line 15) and Study design (line 71). Test-retest designs are usually applied to investigate the agreement or reliability of measures. In this case it appears that a before-and-after (quasi-experimental) design was applied. The difference is that time matters in the latter.
2. Methods. Given comment #1, the manuscript could benefit from specific reporting guidelines available at the EQUATOR Network. After reading Table 2 and its description, it become more apparent that this is a before-and-after (quasi-experimental) design. As such, there is important information that should be disclosed (e.g., see the TIDieR, and others).
3. Methods, lines 189-190. Statistical analysis is possible even for n=1 trials. Indeed, a descriptive statistical analysis was conducted as mentioned in lines 206-207. Consider rephrasing or deleting this sentence.
Minor comments
1. Methods, lines 105-106. Suggestion for future studies is usually reported in Discussion and/or Conclusions.
2. Table 3. Try to keep the last line of this table in the same page of the entire table.
3. Tables 3-5. Include some measure of dispersion (e.g., SD? SE?) alongside the average values.
4. There is no need to overstate this is a ‘proof-of-concept’ design throughout the manuscript.
Author Response
Thank you for reviewer our paper and your valuable recommendations. We have attempted to address all of your concerns and answer all questions.

Round 2
Reviewer 2 Report
Thank you for the opportunity to discuss your study. All comments were properly addressed. I have no new comments.